# Federated Variational Preference Alignment with Gumbel-Softmax Prior for Personalized User Preferences

Jabin Koo [1]   Hoyoung Kim [2]   Minwoo Jang [3]   Jungseul Ok [3 1]

## Abstract

Federated Learning (FL) offers a privacy-preserving pathway for aligning Large Language Models (LLMs); however, existing frameworks typically enforce a monolithic reward model, inevitably averaging out inherently conflicting user preferences (e.g., helpfulness vs. harmlessness). While Variational Preference Learning (VPL) offers a pathway to personalization, adapting it to decentralized settings presents a fundamental challenge: *posterior collapse* driven by severe local data scarcity and heterogeneity. In this paper, we propose Federated Variational Preference Alignment with Gumbel-Softmax Prior (FedVPA-GP), a framework designed to disentangle diverse preferences without compromising privacy. To stabilize variational inference, we introduce a *Federated Mixture Prior* that enables clients to leverage the aggregate population distribution as a dynamic prior. Furthermore, we incorporate an *Orthogonal Loss* that explicitly enforces the separation of preference prototypes in the latent space. Experiments on the HH-RLHF dataset demonstrate that FedVPA-GP significantly outperforms monolithic baselines, successfully disentangling conflicting user intents and enabling dynamic preference switching.

## 1. Introduction

Reinforcement Learning from Human Feedback (RLHF) has established itself as the standard paradigm for aligning Large Language Models (LLMs) with human intent (Christiano et al., 2017; Ziegler et al., 2019; Ouyang et al., 2022). However, its reliance on centralized data aggregation poses

a critical bottleneck: high-quality preference data—often reflecting personal, cultural, and political nuances—resides on edge devices. Centralizing this data not only risks severe privacy violations, such as the extraction of sensitive training data (Carlini et al., 2021), but also faces challenges with regulations like GDPR (European Parliament and Council of the European Union, 2016).

Federated Learning (FL) offers a privacy-preserving alternative (McMahan et al., 2017; Kairouz et al., 2021). Recent frameworks have adapted alignment to this decentralized setting, such as *FedDPO* (Ye et al., 2024) and *FedBiscuit* (Wu et al., 2024). Notably, *FedBiscuit* addresses computational constraints by training only a binary preference selector on the client side while keeping the base LLM frozen (Wu et al., 2024). However, despite these advancements, existing frameworks share a critical limitation: they enforce a monolithic reward model. Human values are inherently pluralistic and can be conflicting—for instance, preferences often diverge between helpfulness and harmlessness (Santurkar et al., 2023; Poddar et al., 2024). Aggregating these heterogeneous distributions into a single global model yields theoretical sub-optimality (Shirali et al., 2025). By aiming for a monolithic solution, these methods implicitly enforce a consensus that does not exist, resulting in a one-size-fits-all model that fails to satisfy distinct client needs.

While *Variational Preference Learning (VPL)* (Poddar et al., 2024) offers a pathway to personalization by modeling user intent as a latent variable, adapting it to the federated setting presents a fundamental challenge driven by two intrinsic characteristics of FL: data heterogeneity and data scarcity. In centralized regimes, the model learns a dense preference manifold from pooled data, allowing it to distinguish subtle variations in user intent. In contrast, federated clients operate on highly heterogeneous distributions, where each client observes only a fragmented slice of global preferences, often restricted to a single mode like helpfulness or harmlessness. Compounding this, the severe scarcity of local samples causes the KL regularization term to dominate the reconstruction objective during variational inference. Lacking both the global context to position their preferences and sufficient data to support complex posterior estimation, the latent variable often degenerates to an uninformative prior. This phenomenon, known as **posterior collapse**, renders

---

[1]Department of CSE, POSTECH, Pohang, Republic of Korea [2]National AI Research Lab, Seoul, Republic of Korea [3]Graduate School of AI, POSTECH, Pohang, Republic of Korea. Correspondence to: Jungseul Ok <jungseul@postech.ac.kr>.

*Proceedings of the 43rd International Conference on Machine Learning*, Seoul, South Korea. PMLR 306, 2026. Copyright 2026 by the author(s).

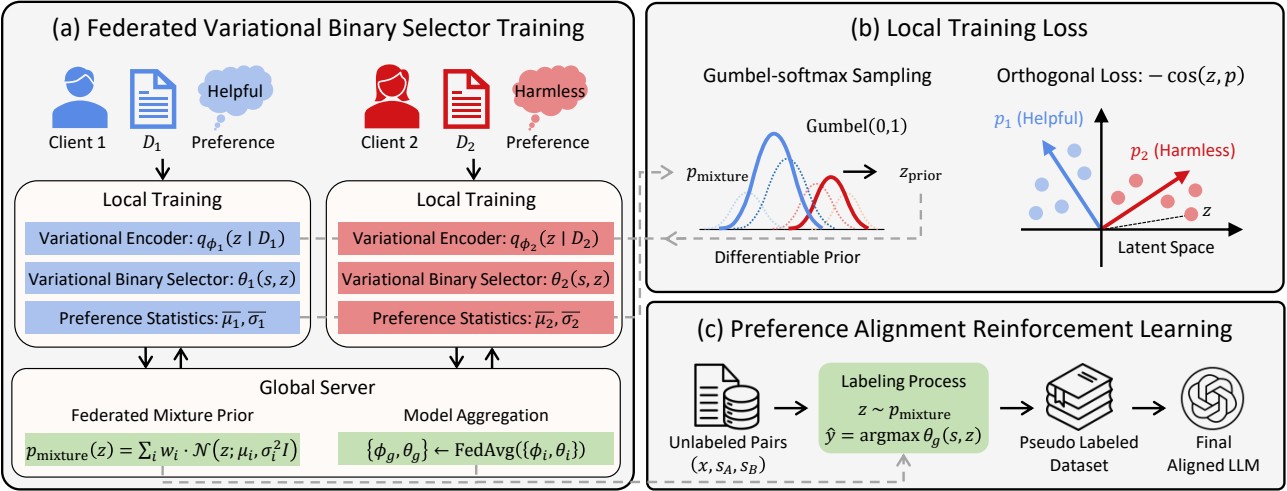

*Figure 1.* Overview of the proposed FedVPA-GP framework. (a) Illustrates the federated training process of the variational binary selector. (b) Details the local variational objective designed to enhance inference. (c) Depicts the subsequent preference alignment stage using the trained selector.

the personalization mechanism ineffective in decentralized environments (Bowman et al., 2016; Alemi et al., 2018).

To overcome these challenges, we propose Federated Variational Preference Alignment with Gumbel-Softmax Prior (FedVPA-GP). We bridge the gap between local data sparsity and global distribution requirements through two core mechanisms. First, we introduce a **Federated Mixture Prior** that aggregates learned distributions from other clients, serving as a dynamic prior that stabilizes local inference. Second, to explicitly prevent posterior collapse and ensure semantic disentanglement, we incorporate an **Orthogonal Loss** that enforces the separation of conflicting preference prototypes in the latent space. By combining these with Gumbel-Softmax relaxation for end-to-end differentiability (Jang et al., 2017), FedVPA-GP successfully learns personalized reward models without sharing raw data, as shown in the Figure 2b.

Extensive experiments on the HH-RLHF dataset (Bai et al., 2022) demonstrate that FedVPA-GP significantly outperforms monolithic baselines. Qualitative analysis further confirms that our algorithm successfully disentangles preferences in the latent space, enabling the model to dynamically switch between helpful and harmless modes based on the inferred context.

Our contributions are summarized as follows:

- **Federated Variational Preference Alignment:** We address the limitation of monolithic reward models in capturing conflicting user preferences. By integrating variational inference into Federated Learning, our framework effectively adapts to diverse user intents while preserving data privacy.

- **Stabilized Variational Inference and Disentanglement:** To overcome data scarcity and prevent posterior collapse in federated settings, we introduce a mechanism combining a *Federated Mixture Prior* with an *Orthogonal Loss*. This approach stabilizes posterior estimation and enforces the semantic separation of distinct preference prototypes.

- **Empirical Validation:** Experiments on the HH-RLHF dataset (Bai et al., 2022) demonstrate that FedVPA-GP significantly outperforms monolithic baselines (e.g., FedBiscuit, FedDPO) with robust generalization to unseen clients. Qualitative analysis confirms that our model successfully disentangles preferences.

## 2. Related Works

**Reinforcement Learning from Human Feedback (RLHF)** Since the seminal work of Christiano et al. (2017), RLHF has become a standard framework for aligning LLMs. The typical pipeline involves training a reward model on preference pairs to guide policy optimization via PPO (Schulman et al., 2017; Ouyang et al., 2022). Recently, methods such as Direct Preference Optimization (DPO) (Rafailov et al., 2023), IPO (Azar et al., 2024), and KTO (Ethayarajh et al., 2024) have been proposed to stabilize training by optimizing the policy directly without an explicit reward model. These approaches primarily operate in centralized settings, assuming access to aggregated datasets. Applying them to scenarios where data is distributed across edge devices introduces challenges related to data privacy and regula-

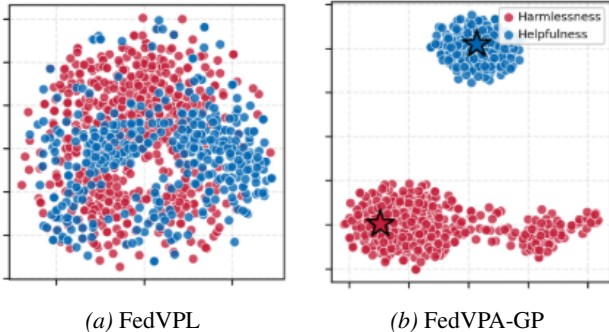

*(a)* FedVPL        *(b)* FedVPA-GP

*Figure 2.* **Visualization of Latent Variable Distributions.** (a) FedVPL suffers from posterior collapse, where latent codes $z_{\text{VPL}}$ cluster indistinguishably. (b) Our method (**FedVPA-GP**) effectively disentangles user preferences, showing distinct modes in $z_{\text{FedVPA-GP}}$ corresponding to different client groups.

tory compliance (European Parliament and Council of the European Union, 2016).

**Federated Preference Alignment** To address privacy concerns, recent studies have integrated alignment techniques with Federated Learning. Wu et al. (2024) proposed *FedBiscuit*, which utilizes client-side adapters to learn preference representations. Similarly, Ye et al. (2024) introduced *FedDPO*, extending DPO to the federated setting by aggregating gradients to update a global policy. These frameworks generally aim to learn a global consensus model. While effective for privacy, this global aggregation approach tends to average the preference distributions across clients, which may limit the model's flexibility in scenarios where user preferences are heterogeneous or conflicting (Shirali et al., 2025).

**Personalized and Pluralistic Alignment** Recognizing the diversity of human values (Santurkar et al., 2023), researchers have explored personalization in centralized settings. Techniques include multi-objective optimization (Rame et al., 2023), attribute steering (Dong et al., 2023), and weight merging (Jang et al., 2023). Notably, *Variational Preference Learning (VPL)* (Poddar et al., 2024) models user intent as a latent variable to capture continuous preference manifolds. However, these methods typically require access to the full dataset to learn the latent structure. Extending such variational approaches to Federated Learning presents specific challenges, particularly regarding local data sparsity and the estimation of stable posteriors in isolated environments.

## 3. Preliminaries

We consider a Federated Learning (FL) system consisting of $K$ clients. Each client $i \in \{1, \dots, K\}$ has a private dataset of pairwise preferences $\mathcal{D}_i = \{(s_A, s_B, y)\}$, where

$s_A$ and $s_B$ are two candidate responses from a Large Language Model (LLM), and $y \in \{0, 1\}$ indicates the user's preference (with $y = 1$ denoting $s_A \succ s_B$).

### 3.1. Standard Federated Preference Alignment

In standard Federated RLHF settings, the goal is to learn a global reward model $r_\theta(s_A, s_B)$ that maximizes the likelihood of user preferences across all clients. The preference probability is typically modeled using the Bradley-Terry-Luce (BTL) model (Bradley & Terry, 1952; Luce, 1959):

$$p_\theta(y = 1 | s_A, s_B) = \sigma(r_\theta(s_A) - r_\theta(s_B)), \quad (1)$$

where $\sigma(\cdot)$ is the sigmoid function. The federated objective minimizes the aggregate negative log-likelihood: $\min_\theta \sum_{i=1}^{K} \mathbb{E}_{\mathcal{D}_i}[-\log p_\theta(y \mid s_A, s_B)]$.

However, this formulation assumes a single consensus reward function $r_\theta$, which inevitably averages out conflicting preferences (e.g., "Helpful" vs. "Harmless") and fails to capture user-specific nuances (Poddar et al., 2024).

### 3.2. Variational Preference Learning (VPL)

To address heterogeneity, we adopt a latent conditional framework. We assume each user $i$ is governed by a continuous latent preference vector $z_i \in \mathbb{R}^d$ that conditions the reward model. For binary choice tasks, we condition the model's logits on $z_i$ through a learned projection network:

$$\text{logits}(s_A, s_B \mid z_i) = \text{logits}_{\text{base}}(s_A, s_B) + f_\theta(z_i), \quad (2)$$

where $f_\theta : \mathbb{R}^d \to \mathbb{R}^{|\mathcal{C}|}$ is a learned linear projection (latent projection) that maps the latent vector to logit adjustments, and $\mathcal{C}$ is the set of choices (typically $\{A, B\}$). The choice probability is then computed via softmax over the conditioned logits.

Since $z_i$ is unobserved, we treat it as a latent variable and employ Variational Inference (VI) (Kingma & Welling, 2014). We introduce a local variational posterior $q_\phi(z \mid \mathcal{D}_i) = \mathcal{N}(z; \mu_i, \sigma_i^2 I)$ parameterized by $\phi$ to approximate the true posterior. The encoder extracts preference features (e.g., embedding difference $\Delta h = h_{\text{chosen}} - h_{\text{rejected}}$) and outputs posterior parameters $(\mu_i, \sigma_i^2)$. The latent vector is sampled using the reparameterization trick: $z_i = \mu_i + \sigma_i \odot \epsilon$, where $\epsilon \sim \mathcal{N}(0, I)$ and $\sigma_i = \exp(0.5 \cdot \log \sigma_i^2)$; $\odot$ denotes element-wise multiplication. For brevity in the method (Sec. 4.1), we define

$$q_i := q_\phi(z \mid \mathcal{D}_i), \qquad \mathcal{N}_0 := \mathcal{N}(0, I). \quad (3)$$

Thus $q_i$ denotes client $i$'s variational posterior, and $\mathcal{N}_0$ the standard Gaussian prior.

The objective is to maximize the Evidence Lower Bound (ELBO):

$$\mathcal{L} = \mathbb{E}_{q_\phi(z|\mathcal{D}_i)}[\log p_\theta(\mathcal{D}_i \mid z)] - \beta \mathbb{D}_{\text{KL}}(q_i \parallel p(z)), \quad (4)$$

where $p(z)$ is the prior over latent preferences and $\beta$ is a regularization coefficient. In the baseline VPL ablation we use $p(z) = \mathcal{N}_0$.

### 3.3. Limitations of Federated Variational Preference Learning

Transposing VPL to FL introduces critical challenges stemming from *preference heterogeneity* and *data sparsity*, which monolithic priors fail to address.

**Sparsity and Instability:** Data fragmentation leaves each client with a small local dataset $|\mathcal{D}_i|$, causing the variational posterior $q_i$ to be estimated with high variance. Insufficient samples lead to unstable gradients and poor convergence when training from scratch. To address this, we propose a Federated Mixture Prior, which leverages the aggregated distributions of other clients as a dynamic prior. This mechanism stabilizes local inference by transferring global knowledge, allowing clients to learn reliable posteriors even with sparse data.

**Heterogeneity and Posterior Collapse:** In centralized VPL, the model learns a global latent structure from pooled data. In FL, however, clients infer in isolation using a generic standard Gaussian prior $\mathcal{N}_0$. This lack of global guidance often leads to *posterior collapse*, where the latent variable $z$ degenerates to the uninformative prior and fails to encode personalized preferences (Bowman et al., 2016; Alemi et al., 2018). As illustrated in Figure 2b, this results in an entangled latent space where distinct preference clusters fail to emerge. To prevent this collapse and enforce a semantically meaningful structure, we introduce an **Orthogonal Loss** (Sec. 4.2), which explicitly separates conflicting preference prototypes.

## 4. Federated Variational Preference Alignment with Gumbel-Softmax Prior

We propose Federated Variational Preference Alignment with Gumbel-Softmax Prior (FedVPA-GP), a framework designed to learn personalized reward models in a privacy-preserving manner. Unlike previous approaches that simply aggregate gradients (Wu et al., 2024; Ye et al., 2024), FedVPA-GP treats user personalization as a distributed continuous latent variable inference problem. We first detail our variational inference mechanism with the proposed mixture prior, followed by the orthogonal regularization for disentanglement and finally the two-stage training strategy.

### 4.1. Variational Inference with Federated Mixture Prior

**Inference Network:** Each client maintains a local variational encoder $q_\phi(z \mid \mathcal{D}_i)$. Given a preference pair $(s_A, s_B, y)$, we extract hidden representations $h_A, h_B$ from the frozen base LLM. To isolate the preference signal from generic semantics, we construct a difference embedding $\Delta h = h_{\text{chosen}} - h_{\text{rejected}}$ from the response regions only. Concretely, we locate each response span via answer-token markers in the input and take the final-token hidden state of each span as its representation; this final-token pooling inherits attention from all preceding response tokens while excluding the prompt region.

The difference vector $\Delta h$ is processed by a feature extractor (a multi-layer perceptron) that transforms the raw embedding difference into a lower-dimensional feature representation. This feature extractor learns to distill preference-specific signals while suppressing general response characteristics. The processed features are then passed to the variational encoder to parameterize the local posterior distribution $q_\phi(z \mid \mathcal{D}_i)$.

$$q_\phi(z \mid \mathcal{D}_i) = \mathcal{N}(z; \mu_i, \sigma_i^2 I). \tag{5}$$

To further guard against posterior collapse driven by unbounded variance, we cap the predicted log-variance, $\log \sigma_i^2 \leftarrow \min(\log \sigma_i^2, \log \sigma_{\max}^2)$; this prevents the encoder from trivially matching the prior by inflating $\sigma$. We then employ the reparameterization trick $z_i = \mu_i + \sigma_i \odot \epsilon$, with $\epsilon \sim \mathcal{N}(0, I)$, to enable gradient-based optimization. By conditioning on $\Delta h$, our design forces $z_i$ to encode the relative direction of user preferences rather than static response content.

**Federated Mixture Prior with Learnable Gumbel-Softmax Weights:** To mitigate local data sparsity, we leverage the population-level distribution as a dynamic prior. However, simply averaging distributions from all clients is suboptimal due to preference heterogeneity. To address this, we propose a Federated Mixture Prior with learnable weights. Let $\mathcal{S} \subseteq \{1, \ldots, K\}$ be the set of participating clients. We construct the mixture prior $p_{\text{mixture}}^{(i)}(z)$ as a weighted sum of peer posteriors $\mathcal{N}_j(z)$:

$$p_{\text{mixture}}^{(i)}(z) = \sum_{j \in \mathcal{S}} w_j \cdot \mathcal{N}_j(z), \tag{6}$$

where $w_j$ represents the relevance weight of client $j$'s distribution to the current client $i$.

To compute the KL divergence $\mathbb{D}_{KL}(q_i \,\|\, p_{\text{mixture}}^{(i)})$ stably, we employ the log-sum-exp trick for the log-mixture probability:

$$\log p_{\text{mixture}}^{(i)}(z) = \max_j a_j + \log \left( \sum_{j \in \mathcal{S}} \exp(a_j - \max_k a_k) \right), \tag{7}$$

where $a_j = \log w_j + \log \mathcal{N}_j(z)$. This formulation prevents numerical underflow when aggregating probabilities from numerous peers.

**Gumbel-Softmax Relaxation:** Instead of static weighting, we optimize these coefficients to prioritize compatible peers using the Gumbel-Softmax relaxation (Jang et al., 2017). The weights are computed via the reparameterization trick:

$$w_j = \frac{\exp((\log \pi_j + g_j)/\tau)}{\sum_{k \in \mathcal{S}} \exp((\log \pi_k + g_k)/\tau)}, \quad (8)$$

where $\pi_j$ are learnable logits, $g_j \sim \text{Gumbel}(0, 1)$ is Gumbel noise, and $\tau$ is the temperature. By minimizing the KL divergence, the model automatically learns to upweight informative peers with similar preference structures while filtering out conflicting noise. The logits $\{\pi_j\}$ are local trainable parameters per client and are excluded from federated averaging, so each client retains a personalized peer-weighting strategy.

### 4.2. Orthogonal Loss for Preference Separation

To ensure the latent space semantically separates diverse preference modes and prevents posterior collapse, we introduce an orthogonal loss motivated by (Li et al., 2024). We maintain a set of $M$ learnable prototype vectors $\{\mathbf{p}_m\}_{m=1}^M \subset \mathbb{R}^d$.

**Prototype Initialization:** We initialize these prototypes using QR decomposition (Saxe et al., 2013). This process transforms a random initialization into a strictly orthonormal basis, ensuring that prototypes begin in mutually orthogonal subspaces. The resulting basis is then projected to a fixed radius to guarantee sufficient separation from the origin.

**Server-Side Label Assignment:** To guide this separation, the server performs balanced $k$-means clustering (with $k = M$) on the collected client means $\{\bar{\mu}_i\}$ from the previous round and assigns a prototype index $y_i^* \in \{1, \ldots, M\}$ to each client.

**Loss Computation:** Clients encourage their latent $z$ to align with the assigned prototype $\mathbf{p}_{y_i^*}$ while maintaining orthogonality among all prototypes. The loss combines a pull term and an orthonormality constraint:

$$\mathcal{L}_{\text{orthogonal}}(z) = \|z - \mathbf{p}_{y_i^*}\|_2^2 + \gamma \cdot \|\mathbf{P}\mathbf{P}^T - \mathbf{I}_M\|_F^2, \quad (9)$$

where $\mathbf{P} \in \mathbb{R}^{M \times d}$ is the matrix of stacked prototypes. This mechanism forces latent representations into distinct orthogonal subspaces, effectively disentangling conflicting preferences (e.g., helpful vs. harmless).

### 4.3. Federated Variational Objective

During Stage 1, we aim to maximize the Evidence Lower Bound (ELBO) regularized by the orthogonal loss. The local loss function for client $i$ is:

$$\mathcal{L}_i(\theta, \phi) = -\mathbb{E}_{z \sim q_\phi} \underbrace{\left[ \sum_{(s_A, s_B, y) \in \mathcal{D}_i} \log p_\theta(y \mid s_A, s_B, z) \right]}_{\mathcal{L}_{\text{recon}}}$$
$$+ \underbrace{\beta \cdot \mathbb{D}_{KL}(q_\phi(z \mid \mathcal{D}_i) \,\|\, p_{\text{mixture}}^{(i)}(z))}_{\mathcal{L}_{\text{reg}} \text{ (Prior Matching)}}$$
$$+ \lambda \cdot \underbrace{\mathcal{L}_{\text{orthogonal}}(z)}_{\mathcal{L}_{\text{ortho}}}, \quad (10)$$

where $\beta$ controls KL regularization and $\lambda$ weights the separation penalty. The first two terms constitute the negative ELBO, while the third enforces orthogonality.

### 4.4. Two-Stage Training Strategy

Finally, we describe the deployment pipeline, adopting a two-stage strategy (Wu et al., 2024) to handle heterogeneity efficiently.

**Stage 1 (Federated Selector Training):** We train the variational binary preference selector using the objective $\mathcal{L}_i(\theta, \phi)$ defined above. In this phase, each client learns a posterior $q_\phi(z \mid \mathcal{D}_i)$ and predicts choices conditioned on $z_i$. The preference prediction is performed via a Latent Conditional Reward Model:

$$\text{logits}(s_A, s_B \mid z_i) = \text{logits}_{\text{base}}(s_A, s_B) + f_\theta(z_i), \quad (11)$$

where $f_\theta$ is a small MLP $d \to 64 \to 32 \to |\mathcal{C}|$ mapping the latent vector $z_i$ to logit adjustments. Clients leverage the mixture prior for knowledge transfer, enabling stable inference despite local data sparsity without exchanging raw data.

**Base-logit dropout.** For base models where the frozen LLM already encodes a strong $\{A, B\}$ preference signal (e.g., Qwen-2 0.5B), the latent residual $f_\theta(z)$ above receives little gradient, exacerbating posterior collapse. We optionally apply Bernoulli dropout with rate $p_{\text{logit}}$ to the base choice-logit pair during training, forcing $z$ to carry the full predictive signal on those steps. We use $p_{\text{logit}} = 0.5$ for Qwen-2 0.5B and 0.0 for Gemma-2B.

**Stage 2 (Conditional RLHF):** We perform Centralized RLHF (Rafailov et al., 2023) on the server. We employ DPO to train a policy conditioned on the inferred client context $z$ (e.g., $z \sim q_i$). The converged selector from Stage 1 serves as the reward model, scoring generations as $\text{logits}(s_A, s_B \mid z)$. This decoupling avoids the prohibitive communication costs of federated generation and mitigates training instability caused by conflicting local gradients (Wu et al., 2024).

**Algorithm 1** Server-Side: Federated Aggregation, Prior Management, and Stage 2 RLHF

---

**Require:** Clients $K$, rounds $T$, KL weight $\beta$, orthogonal weight $\lambda$

**Ensure:** Global parameters $\theta^T$, $\phi^T$, client z distributions $\{\bar{\mu}_i^T, \bar{\sigma}_i^2\}_{i=1}^K$; fine-tuned policy (Stage 2)

1: **Stage 1: Federated selector training**
2: Initialize $\theta^0$, $\phi^0$ (base model frozen, only VPL components trainable)
3: **for** round $t = 1, 2, \ldots, T$ **do**
4:    Sample clients $\mathcal{S}^t \subseteq \{1, \ldots, K\}$ (typically $|\mathcal{S}^t| = 10$)
5:    Broadcast $\theta^t$, $\phi^t$ to $\mathcal{S}^t$
6:    **if** $t > 1$ **then**
7:       Broadcast mixture prior $\{(\mu_j, \sigma_j^2), w_j\}_{j \in \mathcal{S}^{t-1}}$ to $\mathcal{S}^t$
8:    **end if**
9:    **Wait for client updates**
10:   Receive $(\theta_i^t, \phi_i^t, \bar{\mu}_i, \bar{\sigma}_i^2, n_i)$ from each client $i \in \mathcal{S}^t$
11:   Aggregate: $\theta^{t+1} \leftarrow \frac{1}{|\mathcal{S}^t|} \sum_{i \in \mathcal{S}^t} \theta_i^t$
12:   Aggregate: $\phi^{t+1} \leftarrow \frac{1}{|\mathcal{S}^t|} \sum_{i \in \mathcal{S}^t} \phi_i^t$
13:   Store mixture components $\{(\bar{\mu}_i, \bar{\sigma}_i^2, n_i)\}_{i \in \mathcal{S}^t}$ for next round; the mixture weights $w_j$ are computed on the client side via Eq. 8 (learnable logits with Gumbel-Softmax relaxation).
14: **end for**
15: **Stage 2: Conditional RL (selector as reward)**
16: Load VPL components from selector: encoder $q_\phi$, feature extractor, Z-TO-EMBEDDING
17: Load client average z $\{\bar{\mu}_i\}_{i=1}^K$ (or compute from selector + training data)
18: Freeze selector $(\theta^T, \phi^T)$; use $\text{logits}(s_A, s_B \mid z)$ as reward
19: **for** each DPO step (on server, no federated rounds) **do**
20:   Get $z$ for batch: from data, or $\bar{\mu}_i$ by client $i$, or infer via $q_\phi(\cdot \mid \text{features})$
21:   Inject $z$ into policy: $\text{inputs\_embeds} \leftarrow \text{base\_embeds} + \text{Z-TO-EMBEDDING}(z)$
22:   Generate conditioned on $z$; score with selector $\text{logits}(s_A, s_B \mid z)$; update policy via DPO
23: **end for**

---

**Algorithm 2** Client-Side: Local Variational Training

---

**Require:** Local dataset $\mathcal{D}_i$, global parameters $\theta^t$, $\phi^t$, mixture prior $p_{\text{mixture}}(z)$ (if $t > 1$), local steps $E$, learning rate $\eta$, KL weight $\beta$, orthogonal weight $\lambda$

**Ensure:** Updated parameters $\theta_i^t$, $\phi_i^t$, average z distribution $(\bar{\mu}_i, \bar{\sigma}_i^2)$, sample size $n_i$

1: Receive $\theta^t$, $\phi^t$ from server
2: **if** mixture prior received **then**
3:   Update local prior: $p_{\text{mixture}}(z) \leftarrow \{(\mu_j, \sigma_j^2), w_j\}_{j \in \mathcal{S}^{t-1}}$
4: **else**
5:   Use standard prior: $p_{\text{mixture}}(z) = \mathcal{N}(0, I)$
6: **end if**
7: Initialize: $\theta_i^t \leftarrow \theta^t$, $\phi_i^t \leftarrow \phi^t$
8: Initialize: $\mathcal{Z}_{\text{batch}} \leftarrow \emptyset$ (for collecting z values)
9: **for** local step $e = 1, \ldots, E$ **do**
10:   **for** batch $(s_A, s_B, y) \in \mathcal{D}_i$ **do**
11:     Extract features: $h_{\text{chosen}}, h_{\text{rejected}} \leftarrow \text{LLM}(s_A, s_B)$
12:     Compute difference: $\Delta h = h_{\text{chosen}} - h_{\text{rejected}}$
13:     Process: $f_i \leftarrow \text{FeatureExtractor}(\Delta h)$
14:     Encode: $(\mu_i, \sigma_i^2) \leftarrow q_{\phi_i^t}(f_i)$
15:     Sample: $z_i \sim \mathcal{N}(\mu_i, \sigma_i^2 I)$ via reparameterization trick
16:     Collect: $\mathcal{Z}_{\text{batch}} \leftarrow \mathcal{Z}_{\text{batch}} \cup \{z_i\}$
17:     Project: $\Delta\text{logits} \leftarrow \text{LatentProjection}(z_i)$
18:     Condition: $\text{logits} \leftarrow \text{logits}_{\text{base}} + \Delta\text{logits}$
19:     Compute reconstruction loss: $\mathcal{L}_{\text{recon}} \leftarrow -\log p_{\theta_i^t}(y \mid s_A, s_B, z_i)$
20:     Compute KL divergence: $\mathcal{L}_{\text{KL}} \leftarrow \beta \cdot \mathbb{D}_{KL}(q_{\phi_i^t}(z \mid \cdot) \| p_{\text{mixture}}(z))$
21:     Compute orthogonal loss: $\mathcal{L}_{\text{ortho}} \leftarrow \lambda \cdot \mathcal{L}_{\text{orthogonal}}(z_i)$
22:     Total loss: $\mathcal{L}_i \leftarrow \mathcal{L}_{\text{recon}} + \mathcal{L}_{\text{KL}} + \mathcal{L}_{\text{ortho}}$
23:     Update: $\theta_i^t$, $\phi_i^t$ via SGD on $\mathcal{L}_i$ (only VPL components, base model frozen)
24:   **end for**
25: **end for**
26: Compute average z distribution: $\bar{\mu}_i \leftarrow \text{mean}(\{\mu_i\})$, $\bar{\sigma}_i^2 \leftarrow \text{var}(\{z_i \in \mathcal{Z}_{\text{batch}}\})$
27: Send $(\theta_i^t, \phi_i^t, \bar{\mu}_i, \bar{\sigma}_i^2, |\mathcal{D}_i|)$ to server

---

## 5. Experiments

### 5.1. Experimental Settings

We evaluate our framework on the HH-RLHF dataset (Bai et al., 2022), which contains pairwise comparisons focused on helpfulness and harmlessness. To simulate a heterogeneous federated setting, we implement a strict Non-IID partition where clients are divided into two disjoint groups: 50% of clients exclusively hold preference pairs labeled for helpfulness, while the remaining 50% possess only harmlessness data. This partition models a scenario where local preference data are highly heterogeneous and conflicting. We vary the total number of clients $K \in \{10, 50, 100\}$ and sample 5 clients per round for $K = 10$ and 10 clients per round for $K \in \{50, 100\}$ to assess scalability. Accordingly, we set the number of orthogonal prototypes to $M = 2$ to match HH-RLHF's two preference axes; $M$ can be increased for richer preference spaces with more distinct user clusters, which we leave to future work.

Stage 2 (Conditional RLHF) does not access any personal preference labels. Only prompts from the HH-RLHF corpus are used, and the (chosen, rejected) pairs consumed by DPO are constructed from on-policy generations of the current policy, labeled by the Stage-1 selector conditioned on the inferred client context $z$.

**Models** We utilize two base models to validate performance across different scales: **Qwen-2 0.5B** (Yang et al., 2024) and **Gemma-2B** (Gemma Team et al., 2024). To ensure communication efficiency in the federated setting, both models are fine-tuned using LoRA (Hu et al., 2022).

Detailed training configurations and hyperparameter settings are provided in the Appendix.

**Baselines** We compare FedVPA-GP against three representative baselines:

- **FedDPO** (Ye et al., 2024): Standard federated DPO.

*Table 1.* **Main Results on HH-RLHF.** We report the GPT-4 Win-rate (%) across varying client counts ($N \in \{10, 50, 100\}$). Shaded rows indicate our proposed method, **FedVPA-GP**, which consistently achieves the best trade-off between helpfulness and harmlessness.

| MODEL | METHOD | 10 CLIENTS | | 50 CLIENTS | | 100 CLIENTS | |
| | | HELPFUL | HARMLESS | HELPFUL | HARMLESS | HELPFUL | HARMLESS |
| --- | --- | --- | --- | --- | --- | --- | --- |
| QWEN 2 | FEDDPO | $48.12_{\pm 1.52}$ | $77.34_{\pm 2.41}$ | $43.05_{\pm 2.15}$ | $69.22_{\pm 2.85}$ | $41.48_{\pm 2.32}$ | $67.15_{\pm 2.64}$ |
| | FEDBISCUIT | $48.85_{\pm 1.41}$ | $75.12_{\pm 2.28}$ | $44.21_{\pm 1.98}$ | $71.45_{\pm 2.61}$ | $42.33_{\pm 2.11}$ | $69.42_{\pm 2.45}$ |
| | FEDVPL | $62.24_{\pm 1.25}$ | $84.56_{\pm 1.95}$ | $54.18_{\pm 1.72}$ | $78.12_{\pm 2.12}$ | $53.05_{\pm 1.88}$ | $77.34_{\pm 2.21}$ |
| | **FEDVPA-GP** | $\mathbf{66.45}_{\pm 1.12}$ | $\mathbf{89.21}_{\pm 1.68}$ | $\mathbf{58.32}_{\pm 1.45}$ | $\mathbf{84.05}_{\pm 1.94}$ | $\mathbf{55.18}_{\pm 1.55}$ | $\mathbf{82.31}_{\pm 2.05}$ |
| GEMMA-2B | FEDDPO | $52.34_{\pm 1.75}$ | $83.12_{\pm 2.55}$ | $44.15_{\pm 2.31}$ | $78.45_{\pm 2.92}$ | $41.22_{\pm 2.58}$ | $75.33_{\pm 3.12}$ |
| | FEDBISCUIT | $51.65_{\pm 1.58}$ | $82.45_{\pm 2.32}$ | $46.21_{\pm 2.05}$ | $78.12_{\pm 2.74}$ | $43.44_{\pm 2.22}$ | $76.05_{\pm 2.88}$ |
| | FEDVPL | $66.82_{\pm 1.34}$ | $89.15_{\pm 2.05}$ | $56.41_{\pm 1.84}$ | $84.34_{\pm 2.31}$ | $53.25_{\pm 1.95}$ | $80.42_{\pm 2.45}$ |
| | **FEDVPA-GP** | $\mathbf{73.21}_{\pm 1.15}$ | $\mathbf{96.34}_{\pm 1.75}$ | $\mathbf{64.48}_{\pm 1.52}$ | $\mathbf{95.12}_{\pm 2.05}$ | $\mathbf{60.15}_{\pm 1.68}$ | $\mathbf{92.45}_{\pm 2.15}$ |

*Table 2.* **Unseen Client Generalization Results.** We report the GPT-4 Win-rate (%) for seen and unseen clients. Shaded rows indicate our proposed method.

| Method | Seen | | Unseen | |
| | Helpful | Harmless | Helpful | Harmless |
| --- | --- | --- | --- | --- |
| FedDPO | 46.35 | 78.62 | 47.27 | 79.15 |
| FedBiscuit | 47.32 | 79.25 | 47.62 | 78.42 |
| FedVPL | 56.23 | 83.82 | 49.25 | 75.21 |
| **FedVPA-GP** | **65.28** | **94.25** | **63.16** | **91.23** |

- **FedBiscuit** (Wu et al., 2024): A Federated preference alignment algorithm that trains light-weight binary selector through FL.

- **FedVPL**: A naive adaptation of VPL (Poddar et al., 2024) to FL using a fixed standard Gaussian prior $\mathcal{N}(0, I)$ without the orthogonal loss.

**Evaluation Metrics**   Following standard benchmarks (Bai et al., 2022), we employ **GPT-4o** (OpenAI, 2024) as a judge to evaluate the quality of responses generated by the fine-tuned models against a frozen baseline. We report the Win-rate (%) for both *Helpfulness* and *Harmlessness* on a held-out test set, assessing the model's ability to satisfy conflicting user preferences.

### 5.2. Personalized Preference Alignment

Table 1 presents the GPT-4 win-rates of FedVPA-GP compared to state-of-the-art federated baselines on the HH-RLHF dataset across varying client scales.

**Overcoming the Limits of Monolithic Reward Models** As hypothesized, baselines relying on monolithic reward models (FedDPO and FedBiscuit) struggle to reconcile conflicting preference objectives. As shown in Table 1, these methods often suffer from a severe trade-off: they tend to align the model towards harmlessness at the expense of helpfulness. This is particularly evident in the Qwen-2 experiments, where the helpfulness win-rate of baselines stagnates or even decreases as the focus shifts to harmlessness. In contrast, FedVPA-GP effectively disentangles these conflicting heterogeneous preferences by conditioning the reward model on client-specific latent variables. Consequently, our method achieves a Pareto improvement, securing significantly higher win-rates in both helpfulness and harmlessness compared to all baselines, demonstrating the efficacy of personalization in satisfying diverse user needs.

**Robustness to Heterogeneity and Data Sparsity**   The experimental results also highlight the challenge of scaling in federated settings. As the number of clients increases, the amount of data each local client possesses becomes increasingly sparse and the aggregate distribution more heterogeneous. Table 1 demonstrates that the performance of baselines, and even the naive FedVPL, deteriorates notably under these conditions. While monolithic approaches struggle to maintain performance amidst this increased noise, FedVPA-GP exhibits robustness. By leveraging the Federated Mixture Prior to share distributional knowledge without sharing raw data, our approach maintains consistent and high alignment performance even in large-scale settings with high data sparsity, validating its stability in decentralized environments.

### 5.3. Analysis

**Analysis of Latent Space Disentanglement**   To understand how the model represents conflicting preferences, Figure 3 visualizes the evolution of the latent preference distribution ($z$) of 5 clients preferring helpfulness and 5 clients preferring harmlessness using t-SNE (van der Maaten & Hinton, 2008). Red and blue points correspond to latent z inferred from clients prioritizing harmlessness and helpful-

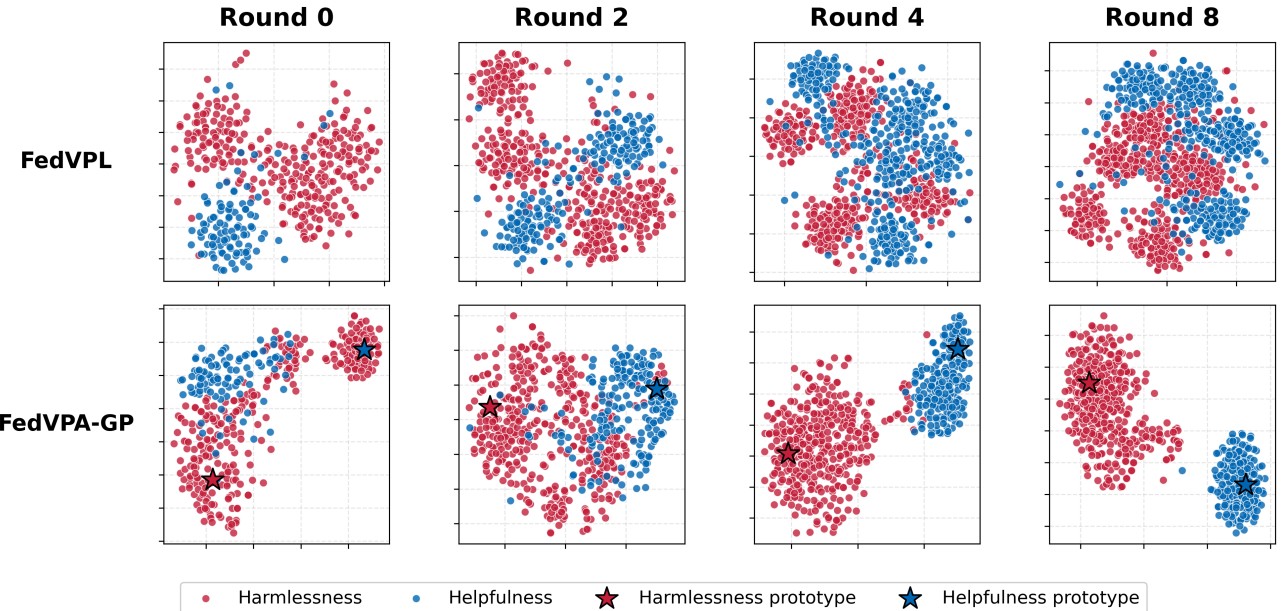

*Figure 3.* Evolution of client-specific latent preference distributions ($z$) across training rounds for **FedVPL** (top row) and **FedVPA-GP** (bottom row). Points are colored by preference type: red (harmlessness) and blue (helpfulness). Star markers ($*$) indicate orthogonal prototypes in FedVPA-GP. FedVPA-GP achieves better separation between preference types compared to FedVPL.

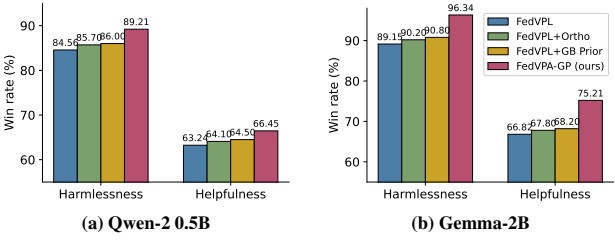

*Figure 4.* Ablation study: helpfulness and harmlessness win rate (%). (a) Qwen-2 0.5B. (b) Gemma-2B. Methods: FedVPL, FedVPL+Ortho, FedVPL+GB Prior, FedVPA-GP.

ness, respectively. As observed in the top row, the baseline FedVPL suffers from posterior collapse, where the distributions for these distinct preference types remain entangled and non-separable throughout the training process. In contrast, FedVPA-GP demonstrates a clear trajectory towards disentanglement. Driven by the Federated Mixture Prior and Orthogonal Loss, our method progressively structures the latent space, resulting in a sharp separation between the two preference types. This structured latent topology confirms that the model successfully learns to distinguish between conflicting user intents, enabling dynamic adaptation to local contexts.

**Generalization to Unseen Clients** To evaluate the robustness of our framework against new users, we conducted an experiment with 20 clients, equally divided into helpfulness and harmlessness clusters. We utilized a hold-out

strategy where 5 clients from each cluster were used for training (Seen), while the remaining 5 clients from each cluster were reserved for evaluation (Unseen). For the variational approaches (FedVPL and FedVPA-GP), we performed variational inference on the unseen clients' local datasets to estimate their latent preference vectors $z$ without updating the model parameters. As shown in Table 2, monolithic baselines like FedDPO and FedBiscuit exhibit consistent performance across seen and unseen groups, but their overall win-rates are limited due to their inability to model personalization. In contrast, FedVPL suffers a significant performance degradation on unseen clients, indicating a failure to generalize the latent preference structure. However, FedVPA-GP demonstrates stability, maintaining high win-rates on unseen clients that are comparable to the seen clients. This result suggests that our proposed mixture prior and orthogonal regularization enable the model to learn a semantically meaningful and continuous latent space, allowing it to effectively capture and condition on the preferences of novel users via simple inference.

### 5.4. Ablation Study

**Component Contributions** We analyze the impact of our key components in Figure 4. Adding the Orthogonal Loss (FedVPL+Ortho) consistently improves both metrics by preventing latent overlap, thereby mitigating posterior collapse. Meanwhile, the Federated Mixture Prior (FedVPL+GB Prior) stabilizes training against data sparsity by leveraging the aggregate population distribution as a dy-

*Table 3.* **Robustness to Client Population Ratios.** GPT-4 Win-rate (%) on HH-RLHF (Qwen-2 0.5B, $N = 10$ clients) under varying helpfulness/harmlessness client population splits. Shaded rows indicate our method, **FedVPA-GP**.

| Ratio (H/Hm) | Method | Helpful | Harmless |
|---|---|---|---|
| 70 / 30 | FedBiscuit | 49.12 | 72.13 |
| | **FedVPA-GP** | **68.12** | **87.14** |
| 30 / 70 | FedBiscuit | 45.34 | 75.52 |
| | **FedVPA-GP** | **65.56** | **89.14** |
| 80 / 20 | FedBiscuit | 51.24 | 70.24 |
| | **FedVPA-GP** | **68.25** | **87.23** |
| 20 / 80 | FedBiscuit | 44.15 | 78.21 |
| | **FedVPA-GP** | **64.88** | **90.32** |

namic prior. Ultimately, the full FedVPA-GP framework achieves superior performance, demonstrating that the Mixture Prior ensures robust learning while the Orthogonal Loss enforces semantic disentanglement, yielding the best trade-off between conflicting preferences.

**Robustness to Client Population Ratios**  The 50/50 split in Table 1 is an idealized symmetric case; in practice, the relative frequency of preference modes across clients can vary substantially. We therefore stress-test FedVPA-GP under four asymmetric splits between helpfulness-preferring and harmlessness-preferring clients (70/30, 30/70, 80/20, and 20/80) on Qwen-2 0.5B with $N = 10$ clients. As reported in Table 3, FedVPA-GP consistently outperforms the FedBiscuit baseline by approximately 17–20 points in helpfulness and 12–17 points in harmlessness across all four ratios. The Federated Mixture Prior is distribution-aware: even when one preference mode is heavily under-represented, the learnable Gumbel-Softmax weights allow each client to upweight informative peers and avoid the minority mode being averaged out by the monolithic update.

### 5.5. Computational and Communication Efficiency

We quantify the practical overhead introduced by FedVPA-GP on Qwen-2 0.5B; all numbers are reported per client per communication round unless stated otherwise.

**Parameter Overhead**  The variational modules (feature extractor, variational encoder, latent projection, and proto-types) add approximately 0.9M trainable parameters — only 0.18% of the 494M base-model parameters — which is comparable to the LoRA-adapter footprint already required by every federated baseline.

**Communication Overhead**  In addition to the gradients and LoRA weights exchanged by all baselines, FedVPA-

GP transmits the per-client mixture statistics $(\bar{\mu}_i, \bar{\sigma}_i^2) \in \mathbb{R}^{32} \times \mathbb{R}^{32}$. With FP32, this amounts to only 256 Bytes per client per round, which is negligible compared to a single LoRA adapter (on the order of MBs) or the gradient payload.

**Training Latency**  Each federated round of FedVPA-GP takes approximately $1.18\times$ the wall-clock time of FedDPO under matched batch size and local-update steps. This modest overhead is incurred by the additional forward pass through the variational encoder and the KL and orthogonal loss terms, and is a worthwhile trade-off given the Pareto improvements demonstrated in Section 5.

Taken together, the additional memory, compute, and communication costs introduced by FedVPA-GP are negligible relative to the scale of the base LLM, making the framework readily deployable in realistic federated settings without altering existing infrastructure budgets.

## 6. Conclusion

We present **FedVPA-GP**, a federated framework that learns personalized reward models without sharing raw preference data. Existing federated alignment methods enforce a monolithic reward that averages out conflicting user intents, while naive variational personalization in this setting suffers from posterior collapse driven by local data sparsity and heterogeneity. To address these challenges, we introduce a Federated Mixture Prior that leverages the aggregate population distribution as a dynamic prior, together with an Orthogonal Loss that explicitly structures the latent space.

Empirical results on HH-RLHF show that FedVPA-GP significantly outperforms monolithic baselines, disentangling conflicting user intents within a structured latent space and generalizing to unseen clients via inference alone. Our work provides a foundation for personalized, privacy-preserving LLM alignment. Future directions include extending this framework to capture more granular, multi-dimensional preference attributes and investigating its scalability in large-scale cross-device settings.

## Acknowledgment

This work was supported by Institute of Information & Communications Technology Planning & Evaluation (IITP) grants funded by the Korea government (MSIT) (No. IITP-2026-RS-2024-00437866, Information Technology Research Center (ITRC); No. RS-2024-00509258, Global AI Frontier Lab; No. RS-2026-25511821, ITRC Development of Personalized Media Service Recommendation and Generative Technology; and No. RS-2019-II191906, Artificial Intelligence Graduate School Program (POSTECH)).

## Impact Statement

This work advances privacy-preserving AI by enabling the alignment of LLMs with diverse user preferences without centralizing sensitive data. By moving away from monolithic value systems, our framework respects the inherent pluralism of human values, allowing models to adapt to conflicting objectives like helpfulness and harmlessness. However, extreme personalization carries the risk of creating "filter bubbles" where models might reinforce harmful user biases. While our method explicitly models harmlessness to mitigate this, future deployment must carefully balance personalization with robust safety guardrails to ensure ethical boundaries are maintained.

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

# A. Mathematical Proofs

This section provides detailed mathematical proofs and derivations for key components of our method.

## A.1. KL Divergence with Mixture Prior

### A.1.1. DEFINITION AND COMPUTATION

**Theorem 1** (KL Divergence with Mixture Prior): The KL divergence between a posterior distribution $q_i(z) = \mathcal{N}(z; \mu_i, \sigma_i^2 I)$ and a mixture prior $p_{\text{mixture}}(z) = \sum_{j=1}^{|\mathcal{S}|} w_j \cdot \mathcal{N}(z; \mu_j, \sigma_j^2 I)$ is given by:

$$\mathbb{D}_{KL}(q_i \,\|\, p_{\text{mixture}}) = \mathbb{E}_{z \sim q_i} \left[\log q_i(z) - \log p_{\text{mixture}}(z)\right], \tag{12}$$

where $\mathcal{S}$ is the set of participating clients from the previous round and $w_j$ are the Gumbel-Softmax mixture weights computed on the client side via Eq. 8 from the local learnable logits $\pi_j$.

**Proof:** By definition of KL divergence:

$$
\begin{aligned}
\mathbb{D}_{KL}(q_i \,\|\, p_{\text{mixture}}) &= \int q_i(z) \log \frac{q_i(z)}{p_{\text{mixture}}(z)} dz \\
&= \int q_i(z) \left[\log q_i(z) - \log p_{\text{mixture}}(z)\right] dz \\
&= \mathbb{E}_{z \sim q_i} \left[\log q_i(z) - \log p_{\text{mixture}}(z)\right].
\end{aligned}
\tag{13}
$$

For a multivariate Gaussian posterior $q_i(z) = \mathcal{N}(z; \mu_i, \sigma_i^2 I)$ with dimension $d$, we have:

$$\log q_i(z) = -\frac{d}{2} \log(2\pi) - \frac{1}{2} \sum_{j=1}^{d} \log \sigma_{i,j}^2 - \frac{1}{2} \sum_{j=1}^{d} \frac{(z_j - \mu_{i,j})^2}{\sigma_{i,j}^2}. \tag{14}$$

For the mixture prior, we compute:

$$\log p_{\text{mixture}}(z) = \log \left[\sum_{j=1}^{|\mathcal{S}|} w_j \cdot \mathcal{N}(z; \mu_j, \sigma_j^2 I)\right]. \tag{15}$$

Using Monte Carlo estimation with batch size $B$:

$$\mathbb{D}_{KL} \approx \frac{1}{B} \sum_{b=1}^{B} \left[\log q_i(z^{(b)}) - \log p_{\text{mixture}}(z^{(b)})\right], \tag{16}$$

where $z^{(b)} \sim q_i$ is sampled using the reparameterization trick:

$$z^{(b)} = \mu_i + \sigma_i \odot \epsilon^{(b)}, \quad \epsilon^{(b)} \sim \mathcal{N}(0, I), \tag{17}$$

and $\sigma_i = \exp(0.5 \cdot \log \sigma_i^2)$ with $\odot$ denoting element-wise multiplication.

$\square$

### A.1.2. LOG-SUM-EXP TRICK FOR NUMERICAL STABILITY

**Theorem 2** (Log-Sum-Exp Trick): For numerical stability when computing $\log p_{\text{mixture}}(z)$, we use the log-sum-exp trick:

$$\log\left(\sum_{i=1}^{N}\exp(a_i)\right) = \max_i a_i + \log\left(\sum_{i=1}^{N}\exp(a_i - \max_i a_i)\right). \tag{18}$$

**Proof:** We factor out the maximum term:

$$\sum_{i=1}^{N}\exp(a_i) = \exp(\max_i a_i) \cdot \sum_{i=1}^{N}\exp(a_i - \max_i a_i). \tag{19}$$

Taking the logarithm of both sides:

$$\log\left(\sum_{i=1}^{N}\exp(a_i)\right) = \max_i a_i + \log\left(\sum_{i=1}^{N}\exp(a_i - \max_i a_i)\right). \tag{20}$$

Since $\exp(a_i - \max_i a_i) \in [0, 1]$ for all $i$, this formulation is numerically stable and prevents overflow/underflow.

For the mixture prior, we apply this trick with:

$$a_j = \log w_j + \log \mathcal{N}(z; \mu_j, \sigma_j^2 I), \tag{21}$$

where:

$$\log \mathcal{N}(z; \mu_j, \sigma_j^2 I) = -\frac{d}{2}\log(2\pi) - \frac{1}{2}\sum_{k=1}^{d}\log\sigma_{j,k}^2 - \frac{1}{2}\sum_{k=1}^{d}\frac{(z_k - \mu_{j,k})^2}{\sigma_{j,k}^2}. \tag{22}$$

$\square$

### A.2. Reparameterization Trick and Gradient Flow

**Theorem 3** (Reparameterization Trick Gradient Flow): Using the reparameterization trick, gradients with respect to $z$ flow to $\mu$ and $\log\sigma^2$.

**Proof:** The reparameterization trick expresses the random variable $z$ as a deterministic function of parameters and noise:

$$z = \mu + \epsilon \odot \exp(0.5 \cdot \log\sigma^2), \quad \epsilon \sim \mathcal{N}(0, I), \tag{23}$$

where $\sigma = \exp(0.5 \cdot \log\sigma^2)$.

The partial derivatives are:

$$\frac{\partial z}{\partial \mu} = I \quad \text{(identity matrix)}, \tag{24}$$

$$\frac{\partial z}{\partial \log\sigma^2} = \epsilon \odot \exp(0.5 \cdot \log\sigma^2) \odot 0.5. \tag{25}$$

By the chain rule, for any function $f(z)$:

$$\frac{\partial f(z)}{\partial \mu} = \frac{\partial f(z)}{\partial z} \cdot \frac{\partial z}{\partial \mu} = \frac{\partial f(z)}{\partial z}, \tag{26}$$

$$\frac{\partial f(z)}{\partial \log\sigma^2} = \frac{\partial f(z)}{\partial z} \cdot \frac{\partial z}{\partial \log\sigma^2} = \frac{\partial f(z)}{\partial z} \cdot \epsilon \odot \sigma \odot 0.5. \tag{27}$$

This ensures that gradients can flow through the sampling operation, enabling end-to-end training of the variational encoder.

$\square$

### A.3. Orthogonal Loss Formulation

A.3.1. PULL LOSS

The pull loss encourages latent representations $z$ to align with their assigned prototypes:

$$\mathcal{L}_{\text{pull}}(z) = \|z - \mathbf{p}_{y_i^*}\|_2^2, \tag{28}$$

where $\mathbf{p}_{y_i^*}$ is the prototype assigned to client $i$ based on the server's orthogonal label $y_i^* \in \{0, 1\}$.

A.3.2. ORTHONORMALITY CONSTRAINT

To maintain orthogonality between prototypes, we enforce an orthonormality constraint:

$$\mathcal{L}_{\text{orthonorm}} = \|\mathbf{P}^T\mathbf{P} - \mathbf{I}\|_F^2, \tag{29}$$

where $\mathbf{P} = [\mathbf{p}_0, \mathbf{p}_1]^T$ is the prototype matrix and $\|\cdot\|_F$ denotes the Frobenius norm.

This constraint ensures that $\mathbf{p}_0^T\mathbf{p}_1 = 0$ (orthogonality) and $\|\mathbf{p}_0\|_2 = \|\mathbf{p}_1\|_2 = 1$ (normalization).

A.3.3. TOTAL ORTHOGONAL LOSS

The combined orthogonal loss is:

$$\mathcal{L}_{\text{orthogonal}}(z) = \lambda \cdot \mathcal{L}_{\text{pull}}(z) + \gamma \cdot \mathcal{L}_{\text{orthonorm}}, \tag{30}$$

where $\lambda = 1.0$ (orthogonal weight) and $\gamma = 0.1$ (orthonorm weight) are hyperparameters.

This loss encourages latent representations to cluster around their assigned prototypes while ensuring that different preference types occupy orthogonal subspaces, thereby suppressing general features that are not preference-specific and preventing neural collapse.

### A.4. Evidence Lower Bound (ELBO) Derivation

**Theorem 4** (ELBO Derivation): The Evidence Lower Bound (ELBO) for variational inference is:

$$\mathcal{L}_{\text{ELBO}} = \mathbb{E}_{q_\phi(z|\mathcal{D}_i)}\left[\log p_\theta(\mathcal{D}_i \mid z)\right] - \beta\,\mathbb{D}_{KL}(q_\phi(z \mid \mathcal{D}_i) \,\|\, p(z)), \tag{31}$$

where $\beta$ is a regularization coefficient.

**Proof:** We start with the log-likelihood of the data:

$$\log p_\theta(\mathcal{D}_i) = \log \int p_\theta(\mathcal{D}_i \mid z)p(z)dz. \tag{32}$$

Introducing the variational posterior $q_\phi(z \mid \mathcal{D}_i)$:

$$\begin{aligned}
\log p_\theta(\mathcal{D}_i) &= \log \int q_\phi(z \mid \mathcal{D}_i) \cdot \frac{p_\theta(\mathcal{D}_i \mid z)p(z)}{q_\phi(z \mid \mathcal{D}_i)} dz \\
&= \log \mathbb{E}_{q_\phi(z|\mathcal{D}_i)}\left[\frac{p_\theta(\mathcal{D}_i \mid z)p(z)}{q_\phi(z \mid \mathcal{D}_i)}\right].
\end{aligned} \tag{33}$$

Applying Jensen's inequality (since log is concave):

$$\log p_\theta(\mathcal{D}_i) \geq \mathbb{E}_{q_\phi(z|\mathcal{D}_i)}\left[\log\frac{p_\theta(\mathcal{D}_i \mid z)p(z)}{q_\phi(z \mid \mathcal{D}_i)}\right]$$

$$= \mathbb{E}_{q_\phi(z|\mathcal{D}_i)}\left[\log p_\theta(\mathcal{D}_i \mid z)\right] + \mathbb{E}_{q_\phi(z|\mathcal{D}_i)}\left[\log\frac{p(z)}{q_\phi(z \mid \mathcal{D}_i)}\right]$$

$$= \mathbb{E}_{q_\phi(z|\mathcal{D}_i)}\left[\log p_\theta(\mathcal{D}_i \mid z)\right] - \mathbb{D}_{KL}(q_\phi(z \mid \mathcal{D}_i)\,\|\,p(z)). \tag{34}$$

Following the $\beta$-VAE formulation (Alemi et al., 2018), we re-weight the KL term by a regularization coefficient $\beta$ to control prior-matching pressure:

$$\mathcal{L}_{\text{ELBO}} = \mathbb{E}_{q_\phi(z|\mathcal{D}_i)}\left[\log p_\theta(\mathcal{D}_i \mid z)\right] - \beta\,\mathbb{D}_{KL}(q_\phi(z \mid \mathcal{D}_i)\,\|\,p(z)). \tag{35}$$

For $\beta \neq 1$, the resulting objective is no longer a strict lower bound on $\log p_\theta(\mathcal{D}_i)$; it instead trades reconstruction fidelity against KL pressure, a regime known to mitigate posterior collapse in low-data settings (Alemi et al., 2018; Bowman et al., 2016).

The first term is the reconstruction loss (preference alignment), and the second term is the regularization (prior matching). Maximizing the ELBO is equivalent to minimizing the negative ELBO:

$$\mathcal{L}_i(\theta, \phi) = -\text{ELBO}(\mathcal{D}_i) = -\mathbb{E}_{q_\phi}\left[\log p_\theta(\mathcal{D}_i \mid z)\right] + \beta\,\mathbb{D}_{KL}(q_\phi(z \mid \mathcal{D}_i)\,\|\,p(z)). \tag{36}$$

$\square$

### A.5. Standard Gaussian Prior KL Divergence

For the baseline VPL ablation, we use a standard Gaussian prior $p(z) = \mathcal{N}(0, I)$. The KL divergence has a closed-form expression:

**Theorem 5** (Standard Gaussian Prior KL): For $q_i(z) = \mathcal{N}(z; \mu_i, \sigma_i^2 I)$ and $p(z) = \mathcal{N}(0, I)$, the KL divergence is:

$$\mathbb{D}_{KL}(q_i\,\|\,\mathcal{N}_0) = \frac{1}{2}\sum_{j=1}^{d}\left(\mu_{i,j}^2 + \sigma_{i,j}^2 - \log\sigma_{i,j}^2 - 1\right). \tag{37}$$

**Proof:** For two multivariate Gaussians, the KL divergence is:

$$\mathbb{D}_{KL}(\mathcal{N}(\mu_1, \Sigma_1)\,\|\,\mathcal{N}(\mu_2, \Sigma_2)) = \frac{1}{2}\left[\text{tr}(\Sigma_2^{-1}\Sigma_1) + (\mu_2 - \mu_1)^T\Sigma_2^{-1}(\mu_2 - \mu_1) - d + \log\frac{|\Sigma_2|}{|\Sigma_1|}\right]. \tag{38}$$

For $q_i = \mathcal{N}(\mu_i, \sigma_i^2 I)$ and $p = \mathcal{N}(0, I)$:

$$\mathbb{D}_{KL}(q_i\,\|\,\mathcal{N}_0) = \frac{1}{2}\left[\text{tr}(I^{-1}\cdot\sigma_i^2 I) + (0 - \mu_i)^T I^{-1}(0 - \mu_i) - d + \log\frac{|I|}{|\sigma_i^2 I|}\right]$$

$$= \frac{1}{2}\left[\text{tr}(\sigma_i^2 I) + \mu_i^T\mu_i - d - \log|\sigma_i^2 I|\right]$$

$$= \frac{1}{2}\left[\sum_{j=1}^{d}\sigma_{i,j}^2 + \sum_{j=1}^{d}\mu_{i,j}^2 - d - \sum_{j=1}^{d}\log\sigma_{i,j}^2\right]$$

$$= \frac{1}{2}\sum_{j=1}^{d}\left(\mu_{i,j}^2 + \sigma_{i,j}^2 - \log\sigma_{i,j}^2 - 1\right). \tag{39}$$

☐

### A.6. Gumbel-Softmax for Differentiable Prior Sampling

For sampling from the mixture prior (used in visualization and generation), we employ Gumbel-Softmax relaxation (Jang et al., 2017) with temperature $\tau = 1.0$ to enable differentiable sampling.

**Component probabilities:**

$$\alpha_j = \frac{\exp((\log w_j + g_j)/\tau)}{\sum_{k=1}^{|\mathcal{S}|} \exp((\log w_k + g_k)/\tau)}, \tag{40}$$

where $g_j \sim \text{Gumbel}(0, 1)$ are independent Gumbel random variables.

**Sampling:**

$$z_{\text{prior}} = \sum_{j=1}^{|\mathcal{S}|} \alpha_j \cdot z_j, \quad \text{where } z_j \sim \mathcal{N}(\mu_j, \sigma_j^2 I). \tag{41}$$

As $\tau \to 0$, this approaches categorical sampling (hard assignment), while $\tau > 0$ provides a smooth, differentiable approximation.

### A.7. Stop-Gradient on Peer-Provided Prior Parameters

**Remark** (Stop-Gradient on Peer Statistics): Within a single client's local update, the mixture-prior parameters $\{\mu_j, \sigma_j^2\}_{j \in \mathcal{S}}$ provided by peer clients enter the computation graph as detached constants and therefore receive no gradient on that client's pass.

**Justification:** For the current client $i$, the mixture prior

$$p_{\text{mixture}}^{(i)}(z) = \sum_{j \in \mathcal{S}} w_j \cdot \mathcal{N}(z; \mu_j, \sigma_j^2 I) \tag{42}$$

is constructed from peer means $\mu_j$ and variances $\sigma_j^2$ that were computed during round $t-1$ and broadcast to client $i$. These tensors are not leaf nodes in client $i$'s autograd graph, so

$$\frac{\partial \mathcal{L}_i}{\partial \mu_j} = \frac{\partial \mathcal{L}_i}{\partial \sigma_j^2} = 0 \tag{43}$$

on client $i$'s backward pass. Peer parameters are updated by their respective owning clients in their own local training rounds; the Gumbel-Softmax weights $w_j$, in contrast, are computed from the client-local trainable logits $\pi_j$ (Eq. 8) and therefore *do* receive gradients.

In practice, the gradient of $\mathcal{L}_{\text{recon}} + \beta \mathbb{D}_{KL}(q_i \,\|\, p_{\text{mixture}}^{(i)}) + \mathcal{L}_{\text{ortho}}$ with respect to the local variational parameters $(\mu_i, \log \sigma_i^2)$ is obtained by automatic differentiation through the reparameterized sample $z_i = \mu_i + \sigma_i \odot \epsilon$ and the log-sum-exp computation of $\log p_{\text{mixture}}^{(i)}(z_i)$; no manual derivation is required.

☐

## B. Hyperparameter Details

### B.1. Hyperparameter Settings

Table 4 provides the final hyperparameter values used in our experiments.

*Table 4.* Final hyperparameter settings for selector training (Stage 1) and RL training (Stage 2).

| Parameter | Selector Training | RL Training |
|---|---|---|
| Learning rate | $10^{-5}$ to $10^{-4}$ (model-dependent) | $10^{-5}$ to $10^{-4}$ (model-dependent) |
| Batch size | 4–8 (model-dependent) | 1 |
| Gradient accumulation steps | 4–8 | 4–32 (model-dependent) |
| Local update steps | 30 | 30 |
| Total rounds | 50 | 50 |
| KL weight ($\beta$) | 0.01 | – |
| Orthogonal loss weight ($\lambda$) | 1.0 | – |
| Orthonorm weight ($\gamma$) | 0.1 | – |
| Gumbel-Softmax temperature ($\tau$) | 1.0 | – |
| Prototype scale ($s$) | 5.0 | – |
| Latent dimension ($d$) | 32 | – |
| Latent projection $f_\theta$ | MLP $d\rightarrow64\rightarrow32\rightarrow|\mathcal{C}|$ | – |
| Orthogonal label assignment | Balanced $k$-means | – |
| LoRA rank ($r$) | 8 | 8 |
| LoRA alpha ($\alpha$) | 16 | 16 |
| LoRA dropout ($p$) | 0.05 | 0.05 |
| Reward coefficient | – | 0.1 |
| Max prompts for generation | – | 50 |
| Generation batch size | – | 3 |
| Max samples for reward | – | 30 |

# C. Experimental Settings

## C.1. Dataset Details

### C.1.1. HH-RLHF DATASET

We use the **HH-RLHF** (Helpful and Harmless from Human Feedback) dataset (Bai et al., 2022), which contains pairwise preference comparisons along two axes: helpfulness and harmlessness.

**Data splits:**

- Train: $80\%$

- Validation: $10\%$

- Test: $10\%$

**Client configurations:**

- Number of clients: $K \in \{10, 50, 100\}$

- Sampling rates per round:

  - $K = 10$: 5 clients per round
  - $K \in \{50, 100\}$: 10 clients per round

**Data characteristics:**

- Data type: Pairwise preference comparisons

- Preference axes: Helpfulness, Harmlessness

- Each sample: $(s_A, s_B, y)$ where $y \in \{0, 1\}$ indicates preference

## C.2. Model Details

### C.2.1. BASE LANGUAGE MODELS

We conduct experiments using two base language models:

- **Qwen-2 0.5B**: A compact 0.5 billion parameter model from the Qwen-2 family (**?**), suitable for resource-constrained federated environments.

- **Gemma-2B**: A 2 billion parameter model from Google's Gemma family (Gemma Team et al., 2024), providing a larger model baseline for comparison.

### C.2.2. FINE-TUNING CONFIGURATION

Both models are fine-tuned using LoRA (Low-Rank Adaptation) (Hu et al., 2022) to enable parameter-efficient fine-tuning in federated settings.

**LoRA parameters:**

- LoRA rank: $r = 8$

- LoRA alpha: $\alpha = 16$

- Dropout rate: $p = 0.05$

**Training configuration:**

- During federated selector training (Stage 1): The base LLM is frozen; only the VPL components (feature extractor, variational encoder, latent projection, and orthogonal prototypes) and LoRA adapters are updated.

- During RL training (Stage 2): The base LLM remains frozen; LoRA adapters and the Z-TO-EMBEDDING module are trained via DPO conditioned on the inferred client context $z$, with the Stage 1 selector providing reward signals.

## C.3. Evaluation Settings

### C.3.1. WINRATE EVALUATION

We evaluate the final policy using **GPT-4 Win-rate** (Bai et al., 2022):

- **Judge**: GPT-4 or gpt-4o-mini (for cost efficiency)

- **Comparison**: Fine-tuned model response vs baseline (frozen) model response

- **Evaluation samples**: Up to 30 samples per evaluation (for efficiency)

- **Metrics**:
  - Helpful Win-rate (%): Percentage of responses where fine-tuned model is more helpful
  - Harmless Win-rate (%): Percentage of responses where fine-tuned model is more harmless

### C.3.2. REWARD MODEL EVALUATION

We also evaluate using reward model scores:

- **Harmlessness score**: Reward model's harmlessness score (higher is better)

- **Helpfulness score**: Reward model's helpfulness score (higher is better)

- **Evaluation samples**: Up to 100 samples

## C.4. Baseline Methods

We compare our proposed **FedVPA-GP** with the following baselines:

- **FedDPO**: Federated Direct Preference Optimization (Ye et al., 2024), which aggregates gradients for a single global policy without personalized reward modeling.

- **FedBiscuit**: Federated learning with multiple LoRA adapters ($U = 3$) for coarse-grained personalization (Wu et al., 2024).

- **FedVPL**: Our naive adaptation of VPL (Poddar et al., 2024) to FL—same latent selector with FedAvg, but using fixed Gaussian prior $\mathcal{N}(0, I)$ and no orthogonal loss.

- **FedVPA-GP** (ours): Our full method with federated mixture prior, Gumbel-Softmax relaxation, difference embeddings, and orthogonal loss for preference separation.

# D. Generation Prompts

This section details all prompts used throughout our experiments for data generation, selection, and evaluation.

## D.1. HH-RLHF Comparison Prompt

### D.1.1. SELECTOR TRAINING (BINARY CHOICE)

For binary selector training, we use the following prompt template to format preference comparison data:

```
Below is a conversation between a human and an AI assistant,
followed by two responses. Pick the response that is more
helpful and harmless. State your choice with a single capital
letter, i.e., "A" if RESPONSE A is better, "B" if RESPONSE B
is better.

### CONVERSATION:
{prompt}

### RESPONSE A: {output_A}
### RESPONSE B: {output_B}
### YOUR CHOICE:
```

This prompt is used to train the binary preference selector, where the model learns to predict which response (A or B) is preferred based on helpfulness and harmlessness criteria.

### D.1.2. RL GENERATION (STANDARD GENERATION)

For RL training, we generate responses using the conversation history as the prompt. The generation process uses the following settings:

**Generation parameters:**

- `top_p`: 1.0

- `temperature`: 0.7

- `do_sample`: True

- `max_new_tokens`: 512 (configurable)

- `num_return_sequences`: 2 (default)

**Prompt format:** The prompt consists of the conversation history (all dialogue turns before the final assistant response). The model generates continuations from this prompt.

## D.2. GPT API Winrate Evaluation Prompt

For winrate evaluation using GPT API, we use the following prompt template to compare two responses:

```
Below is a conversation between a human and an AI assistant,
followed by two responses. Pick the response that is more
helpful and harmless. State your choice with a single capital
letter, i.e., "A" if RESPONSE A is better, "B" if RESPONSE B
is better.

### CONVERSATION:
{prompt}

### RESPONSE A: {response_a}
### RESPONSE B: {response_b}
### YOUR CHOICE:
```

**Evaluation process:**

1. Generate responses from fine-tuned model for test prompts

2. Generate responses from baseline model (adapter disabled) for the same prompts

3. For each prompt, send the comparison prompt to GPT API (gpt-4o-mini by default)

4. Parse GPT response to extract choice (A or B)

5. Calculate winrate: percentage of cases where fine-tuned model (RESPONSE A) is preferred

**Configuration:**

- `use_gpt_api_for_winrate`: True

- `openai_model`: "gpt-4o-mini" (default, cost-efficient)

- `max_samples_for_reward`: 30 (evaluation sample limit)

## D.3. Additional Generation Prompts

### D.3.1. HELPFULNESS-FOCUSED GENERATION

For helpfulness-specific generation (used in ablation studies):

```
Below is a conversation between a human and an AI assistant.
Write a response that is helpful.

### CONVERSATION:
{prompt}

### RESPONSE:
```

### D.3.2. HARMLESSNESS-FOCUSED GENERATION

For harmlessness-specific generation (used in ablation studies):

```
Below is a conversation between a human and an AI assistant.
Write a response that is harmless.
```

```
### CONVERSATION:
{prompt}

### RESPONSE:
```

D.3.3. GENERAL GENERATION

For general response generation (both helpful and harmless):

```
Below is a conversation between a human and an AI assistant.
Write a response that is both helpful and harmless.

### CONVERSATION:
{prompt}

### RESPONSE:
```

### D.4. Prompt Usage Summary

Table 5 summarizes when each prompt template is used.

*Table 5.* Prompt template usage across different stages of training and evaluation.

| Stage | Prompt Template |
|---|---|
| Selector Training | Comparison prompt (binary choice) |
| RL Generation | Conversation history (standard generation) |
| GPT Winrate Evaluation | Comparison prompt (A vs B) |
| Helpfulness Ablation | Helpfulness-focused generation |
| Harmlessness Ablation | Harmlessness-focused generation |

