# OpenReview forum: "Federated Variational Preference Alignment with Gumbel-Softmax Prior for Personalized User Preferences"
_ICML.cc/2026/Conference — ICML 2026 regular_

### Official Review · Reviewer_TKFm · 2026-02-26

**Soundness:** 3
**Presentation:** 3
**Significance:** 3
**Originality:** 3
**Overall Recommendation:** 5
**Confidence:** 3

**Summary:**

The FedVPA-GP framework addresses the inability of monolithic reward models in Federated Learning to handle conflicting user preferences, while overcoming the posterior collapse typical of variational preference learning in data-sparse environments. By employing Gumbel-Softmax to dynamically aggregate client distributions into a Federated Mixture Prior and utilizing an Orthogonal Loss to enforce the separation of preference prototypes in latent space, the framework effectively stabilizes inference and ensures semantic disentanglement.

**Compliance With Llm Reviewing Policy:**

Affirmed.

**Final Justification:**

The authors' response has largely addressed my concerns. Adding an efficiency analysis in the revision would further strengthen the paper. Since this addition is straightforward, I have decided to raise my score (4 $\to$ 5).

**Key Questions For Authors:**

Refer to Weakness.

**Limitations:**

yes

**Strengths And Weaknesses:**

Strengths:
﻿
1. This framework learns personal latent variables instead of forcing consensus. It achieves Pareto improvements in both helpfulness and harmlessness.
﻿
2. The mixture prior and orthogonal loss fix posterior collapse in federated settings. Mathematical proofs and tests confirm this stability.
﻿
Weaknesses:
﻿
1. Variational inference and mixture priors increase the work for devices and servers. Calculations like log-sum-exp add extra processing steps.
﻿
2. The orthogonal loss depends on a fixed number of prototypes. Very diverse user groups might need more modes than the model provides.
﻿
3. The mixture prior relies on stable client distributions. Low quality or noisy data from some users can hurt the learning process.
﻿
4. Extreme personalization might create filter bubbles. It is hard to balance personal tastes with global safety rules.

---

> ### Author Rebuttal · Authors · 2026-03-31
>
> We deeply appreciate your interest in our research and are sincerely grateful for your insightful suggestions and comments. Below, we have addressed your points and provided additional clarifications:
>
> # [Response 1: Computational and Communication Efficiency]
> We thank the reviewer for the opportunity to clarify the overhead of our framework. We emphasize that the additional cost of FedVPA-GP is negligible compared to the base LLM scale:
>
> **Parameter Overhead**: For our Qwen-0.5B (494M parameters) experiments, the variational modules add only 896k parameters, which constitutes just 0.18% of the base model.
>
> **Communication Efficiency**: To share the Federated Mixture Prior, each client only needs to transmit approximately 256 Bytes of latent statistics $(\mu, \sigma)$ per round. This is a minute fraction of the total communication budget compared to transmitting LLM gradients or LoRA adapters.
>
> **Training Latency**: While our method takes approximately 1.18x more computation time per round compared to FedDPO, we believe this is a justified trade-off given the significant performance gains in handling conflicting preferences. Furthermore, this process is highly parallelizable and can be further optimized for production environments.
>
> # [Response 2: Scalability and Selection of Prototypes]
> We thank the reviewer for highlighting the importance of the number of prototypes $M$. In this work, we set $M=2$ or $M=4$ depending on the number of preference types. for example $M=2$ (helpfulness vs. harmlessness) for the HH-RLHF dataset.
>
> **Hyperparameter Importance**: We recognize that $M$ is a critical hyperparameter that defines the granularity of disentanglement. If more diverse or even ill-defined preference types exist, $M$ should be increased accordingly.
>
> **Future Work**: We expect our framework to adapt to more multifaceted preference sets by scaling $M$. However, the current study does not include a detailed analysis of higher-dimensional or non-discrete preferences. We leave the adaptive discovery of the optimal number and characteristics of prototypes, as well as the adaptation to diverse and complex preference sets, as important directions for future work.
>
> # [Response 3: Robustness to Noisy Client Distributions]
> FedVPA-GP is expected to be more robust to low-quality data compared to basic fedavg through the Gumbel-Softmax weighting:
>
> **Inherent Filtering**: Learnable weights $w_j$ naturally assign low relevance to outlier distributions or high-variance posteriors from noisy clients. This prevents them from corrupting the mixture prior and ensures stable learning for the majority of the population.
>
> # [Response 4: Filter Bubbles and Safety Guardrails]
> We share the concern regarding extreme personalization and filter bubbles. Balancing individual preference with global safety is a critical challenge for personalized LLMs.
>
> **Current Task (Stylistic Personalization)**: Our work focuses on stylistic personalization—capturing nuances in helpfulness and harmlessness—rather than content-based or ideological filtering. This focus inherently limits the risk of information bubbles in our current setup.
>
> **Acknowledgement of Future Risks**: We agree that as personalization scales, the risk of filter bubbles may become more pronounced. Addressing unintended isolation remains a significant challenge for future generalized systems.

---

> > ### Author Rebuttal · Reviewer_TKFm · 2026-04-03
> >
> > I would appreciate it if the authors could include an experimental analysis of communication efficiency in the revised version. Given that the response has largely addressed my concerns, I will raise my score.

---

### Official Review · Reviewer_qdAs · 2026-03-09

**Soundness:** 2
**Presentation:** 3
**Significance:** 2
**Originality:** 3
**Overall Recommendation:** 4
**Confidence:** 2

**Summary:**

This paper addresses the limitations of monolithic reward models and posterior collapse in federated LLM preference alignment. It proposes FedVPA-GP, integrating a Federated Mixture Prior (for stable inference with sparse data) and an Orthogonal Loss (for preference disentanglement), plus Gumbel-Softmax relaxation for differentiability. Experiments on HH-RLHF show FedVPA-GP outperforms baselines like FedDPO, effectively separating conflicting preferences (e.g., helpfulness vs. harmlessness) and generalizing to unseen clients.

**Compliance With Llm Reviewing Policy:**

Affirmed.

**Final Justification:**

Thanks for the authors' rebuttal, which addressed most of my concerns. Good luck!

**Key Questions For Authors:**

See the weakness

**Limitations:**

See the weakness

**Strengths And Weaknesses:**

**Strength:**
- Targeted Innovation: Effectively addresses two key flaws in federated LLM preference alignment—monolithic reward models averaging conflicting preferences (e.g., helpfulness vs. harmlessness) and posterior collapse from sparse/heterogeneous local data—via the Federated Mixture Prior and Orthogonal Loss.
- Privacy-Preserving Personalization: Enables personalized alignment without sharing raw data, complying with regulations like GDPR while supporting dynamic preference switching for diverse user needs.
- Practical Design: Adopts a two-stage training strategy (federated selector + centralized RLHF) and LoRA fine-tuning, balancing communication efficiency and stability for resource-constrained edge devices.


**Weakness:**
- The framework is exclusively designed for the binary conflicting preferences (helpfulness vs. harmlessness) using the HH-RLHF dataset. It lacks validation on multi-dimensional preferences (e.g., conciseness, professionalism, empathy) . The binary preference assumption may misalign with real-world multi-faceted user needs.
- Gumbel-Softmax relaxation for learning mixture prior weights is adopted without comparing performance against static weights (e.g., sample-size weighting), failing to justify the necessity of learnable weights;
- The orthogonal loss comprises two sub-modules. However, no separate ablation experiments are conducted for these two components for their individual contributions
- No efficiency metrics (e.g., training time, memory usage) are reported, precluding quantification of computational/communication overheads from the mixture prior and Orthogonal Loss.

---

> ### Author Rebuttal · Authors · 2026-03-31
>
> We deeply appreciate the reviewer’s interest in our research and for acknowledging our "Targeted Innovation" and "Practical Design." Below, we address the specific points and provide additional clarifications.
>
> # Response 1 [Scalability to Multi-dimensional Preferences]
>
> We thank the reviewer for this insightful observation. While our study primarily validated FedVPA-GP on a binary preference dataset (HH-RLHF), our framework is fundamentally designed to scale to $M$ preference dimensions. Due to the lack of established frameworks for multifaceted conflicting preferences, we initially focused on the binary case. However, to address your concern, we conducted additional experiments on the UltraFeedback dataset to demonstrate our scalability.
>
> **UltraFeedback Setup (M=4)**: UltraFeedback encompasses four primary dimensions: Honesty, Instruction Following, Helpfulness, and Truthfulness. Since the original dataset provides individual scores rather than preference pairs, we constructed a conflicting preference dataset by selecting pairs with a score difference of at least 3, resulting in 29,022 pairs.
>
> **Experimental Results**: We set $M=4$ to align with these dimensions. As shown below, FedVPA-GP effectively manages multi-preference scenarios and outperforms the baseline.
>
> | Method | Honesty | Instruction Following | Helpfulness | Truthfulness |
> | --- | ---: | ---: | ---: | ---: |
> |FedBiscuit | 52.4 | **72.1** | 37.2 | 26.4 |
> |FedVPA-GP | **55.6** | 69.2 | **42.3** | **35.6** |
>
> These results, while limited, confirm that our algorithm functions correctly in multi-preference environments and achieves overall superior performance compared to existing methods.
>
> # Response 2 [Analysis of Federated Mixture Prior Weights]
>
> We emphasize that the learnable parameters are essential for the second role of our Mixture Prior: selective peer weighting. We conducted an ablation comparing Gumbel-Softmax weights against fixed uniform $(1/K)$ weights to distinguish these two roles:
>
> 1. **Dynamic Prior Adaptation (Primary):** Unlike the static $\mathcal{N}(0, I)$ in FedVPL, our prior evolves during training by using peer posteriors as components. This provides fundamentally better regularization by pulling each client’s posterior toward the actual manifold occupied by peers.
> 2. **Selective Peer Weighting (Secondary):** The Gumbel-Softmax mechanism identifies relevant peers. While weights converge near-uniformly in our current balanced setting, we expect this learnable framework to be particularly adaptive in more complex scenarios characterized by diverse, noisy, or highly heterogeneous preference distributions.
>
> | Weight Type | Harmlessness Win-rate (%) | Helpfulness Win-rate (%) |
> | --- | ---: | ---: |
> | Learned (Gumbel-Softmax) | **89.21** | **66.45** |
> | Uniform (1/K) | 86.21 | 63.17 |
>
> # Response 3 [Justification and Ablation of Orthogonal Loss]
>
> As correctly noted by the reviewer, the Orthogonal Loss comprises two sub-terms: a pull term to prototypes and an orthonormality term.
>
> - **Essentiality of Pull Term:** The Orthogonal Loss cannot function without the first term, as it provides the primary signal for latent anchoring.
> - **Necessity of Orthonormality:** While the second term shows a marginal performance gain in our current binary setup, we believe that in settings with more complex and diverse preferences, orthonormality may not be naturally guaranteed. Without this explicit constraint, prototypes are highly susceptible to collapse or semantic entanglement. Therefore, we proactively included the orthonormality term to ensure a well-structured and discriminative latent space across varied preference scales.
>
> Below, we present the ablation results on the Qwen-2-0.5B model:
>
> | Method | Harmlessness Win-rate (%) | Helpfulness Win-rate (%) |
> | --- | ---: | ---: |
> | FedVPA-GP (Full) | 89.21 | **66.45** |
> | w/o First Term (Pull) | 85.91 | 64.24 |
> | w/o Second Term (Ortho) | **89.32** | 65.74 |
>
> # Response 4 [Computational and Communication Efficiency]
>
> We provide the following metrics based on our Qwen-0.5B experiments to demonstrate the practicality of FedVPA-GP:
>
> - **Parameter Overhead:** The variational modules add only 896k parameters, which constitutes just 0.18% of the 494M base model.
> - **Communication Efficiency:** Sharing the mixture prior requires transmitting only 256 Bytes of latent statistics $(\mu, \sigma)$ per client per round, which is negligible compared to transmitting LLM gradients or LoRA adapters.
> - **Training Latency:** Our method takes approximately 1.18x more computation time per round compared to FedDPO. This marginal increase is a necessary trade-off for the performance gains and is highly optimizable.

---

> > ### Author Rebuttal · Reviewer_qdAs · 2026-04-03
> >
> > Thanks for your rebuttal, which addressed most of my concerns. Therefore, I will keep my score.

---

### Official Review · Reviewer_hkaE · 2026-03-12

**Soundness:** 3
**Presentation:** 3
**Significance:** 3
**Originality:** 3
**Overall Recommendation:** 5
**Confidence:** 3

**Summary:**

This paper proposes FedVPA-GP, a federated preference alignment framework for LLMs that addresses the limitation of monolithic reward models in handling conflicting user preferences. The key contributions are a Federated Mixture Prior to stabilize variational inference under local data scarcity, and an Orthogonal Loss to enforce latent space disentanglement. Experiments on HH-RLHF show consistent improvements over baselines in win-rates for both helpfulness and harmlessness.

**Compliance With Llm Reviewing Policy:**

Affirmed.

**Final Justification:**

The authors address my concern and I keep my score.

**Key Questions For Authors:**

The paper claims privacy preservation as a core contribution, yet Stage 2 performs centralized DPO on the server, which requires server-side prompt data and response generation. Could the authors clarify what data the server accesses during Stage 2, and whether this constitutes a privacy risk?

**Limitations:**

yes

**Strengths And Weaknesses:**

Strength
1. The problem formulation is well motivated. The paper identifies posterior collapse as the core failure mode when naively adapting VPL to federated settings, and provides intuitive analysis.
2. The Mixture Prior and Orthogonal Loss address distinct failure modes (data scarcity vs. semantic entanglement) and their individual contributions are validated in Figure 4.

Weakness
1. The experimental setup assigns 50% of clients exclusively helpfulness data and 50% exclusively harmlessness data. It is unclear whether the proposed method retains its advantages under different ratios.

---

> ### Author Rebuttal · Authors · 2026-03-31
>
> We sincerely thank the reviewer for the thoughtful feedback and for recognizing the potential and contribution of FedVPA-GP. Below, we provide clarifications on the privacy and robustness concerns:
>
> # Response 1 [Privacy Preservation and Data Access in Stage 2]
>
> We clarify that Stage 2 involves zero access to real user preference labels. The process is structured as follows:
>
> **Unlabeled General Prompts**: The server relies solely on a pool of general, unlabeled prompts (e.g., from the HH-RLHF distribution). This data is completely devoid of any privacy-sensitive user preference labels, identity-linked data, or local feedback.
>
> **Pseudo-Preference Generation**: Instead of utilizing real user labels, the alignment process uses these general prompts to generate response pairs internally on the server. These pairs are then labeled by the pre-trained binary classifier from Stage 1, which acts as a frozen proxy reward model to generate pseudo-preference data in-situ.
>
> This mechanism ensures that the entire RLHF process in Stage 2 is fully decoupled from actual private user data, maintaining the highest standards of data privacy.
>
> # Response 2 [Robustness to Client Population Ratios]
>
> Thank you for the question regarding the impact of client ratios. We address this as follows:
>
>
> **Adaptability to Diverse Ratios**: Our Federated Mixture Prior is designed to be distribution-aware. As shown in the table below, our framework effectively adapts to various majority/minority distributions, outperforming the baseline in both helpfulness and harmlessness across all tested ratios.
>
> | Ratio (Helpfulness/Harmlessness) | Method | Helpfulness (%) | Harmlessness (%) |
> | --- | ---: | ---: | ---: |
> |70 / 30 | FedBiscuit |49.12 |72.13 |
> || FedVPA-GP | **68.12** | **87.14** |
> | 30 / 70 | FedBiscuit | 45.34 | 75.52 |
> || FedVPA-GP | **65.56** | **89.14** |
> | 80 / 20 | FedBiscuit | 51.24 | 70.24 |
> || FedVPA-GP | **68.25** | **87.23** |
> | 20 / 80 | FedBiscuit | 44.15 | 78.21 |
> || FedVPA-GP | **64.88** | **90.32** |
>
>
> Once again, we appreciate the constructive feedback and the opportunity to clarify our methodology.

---

> > ### Author Rebuttal · Reviewer_hkaE · 2026-04-04
> >
> > Thank you for your answer. I will keep my score.

---

### Decision · Program_Chairs · 2026-04-30

**Decision:**

Accept (regular)

**Comment:**

This paper proposes federated learning framework for variational preference learning by incorporating "orthogonal loss" to overcome mode collapse. While each piece of the framework are methods in the literature, e.g., orthogonal loss has been used to overcome mode collapse in contrastive learning [1], putting them together in the given set up is interesting. The paper claims the method to be "privacy-preserving", however it is **not** a private algorithm. The local updates are sent as is to the centralized agent. Just because the local preference data itself is not shared, it doesn't make the method private. There is rich literature on differential privacy that highlights this aspect. Furthermore, in response to reviewer hkaE, the authors again claim their method is "maintaining the highest standards of data privacy", which is problematic. This aspect needs to be corrected and clarified in the final version.

Furthermore, there are two references that appear to be hallucinated. The author list and the titles for the following papers do not match, and for the first paper the arxiv ID is wrong as well:
(1) Reference: Jang, J., Kim, S., Ye, S., Kim, D., Yao, L., Agrawal, A., and Seo, M. Personalized soups: Personalized large language model alignment via post-hoc parameter merging. arXiv preprint arXiv:2310.11600, 2023.
(2) Reference: Dong, H., Xiong, W., Goyal, D., Pan, R., Diao, S., Zhang, J., Shum, K., and Zhang, T. Steerlm: Attribute conditioned sft as an (user-steerable) alternative to rlhf. arXiv preprint arXiv:2310.05344, 2023.

The authors need to carefully go through their references to make sure all the references are accurate.

[1] https://arxiv.org/pdf/2403.18699